# Deciphering Cross-Modal Alignment in Large Vision-Language Models with Modality Integration Rate

## Abstract

We present the Modality Integration Rate (MIR), an effective, robust, and generalized metric to indicate the multi-modal pre-training quality of Large Vision Language Models (LVLMs). Large-scale pre-training plays a critical role in building capable LVLMs, while evaluating its training quality without the costly supervised fine-tuning stage is under-explored. Loss, perplexity, and in-context evaluation results are commonly used pre-training metrics for Large Language Models (LLMs), while we observed that these metrics are less indicative when aligning a well-trained LLM with a new modality. Due to the lack of proper metrics, the research of LVLMs in the critical pre-training stage is hindered greatly, including the training data choice, efficient module design, etc. In this paper, we propose evaluating the pre-training quality from the inter-modal distribution distance perspective and present MIR, the Modality Integration Rate, which is 1) **Effective** to represent the pre-training quality and show a positive relation with the benchmark performance after supervised fine-tuning. 2) **Robust** toward different training/evaluation data. 3) **Generalize** across training configurations and architecture choices. We conduct a series of pre-training experiments to explore the effectiveness of MIR and observe satisfactory results that MIR is indicative about training data selection, training strategy schedule, and model architecture design to get better pre-training results. We hope MIR could be a helpful metric for building capable LVLMs and inspire the following research about modality alignment in different areas.

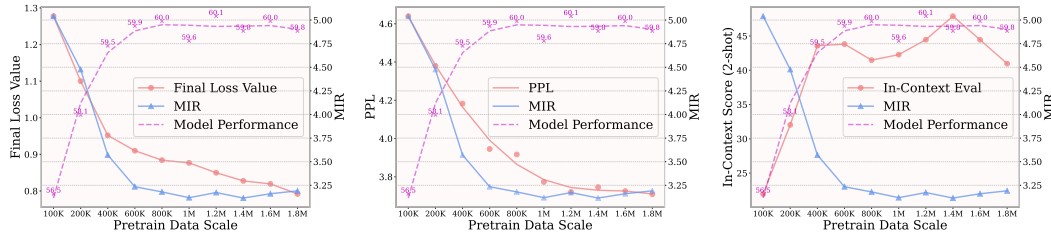

Figure 1: **Final loss, perplexity (PPL), and in-context evaluation are insufficient indicators of LVLM pre-training quality.** We test the effectiveness of these three methods and our proposed MIR on a pre-training data scaling experiment, where we curate ∼1.8M GPT-style data from ALLaVA (Chen et al. (2024a)) and ShareGPT4V-PT (Chen et al. (2023)) and use different amount of data to pre-train LLaVA-1.5 7B models (Liu et al. (2024c)). Note that "Model Performance" means the post-SFT (Supervised Fine-tuning) performance on 7 multi-modal benchmarks after we equally apply SFT on these pre-trained models on LLaVA's 665K SFT data. In (a), we report the average loss over the last 50 pre-training steps as the final loss. In (b), the PPL is calculated on 1,000 randomly sampled image-caption pairs from ShareGPT4V. In (c), we apply 2-shot in-context evaluation and force the pre-trained models to response choice on MME (Fu et al. (2023)), MMBench (Liu et al. (2023)), SEED-Img (Li et al. (2023a)), and report the average scores. We can find that these three metrics fail to measure the pre-training quality while MIR well fits the actual model performance.

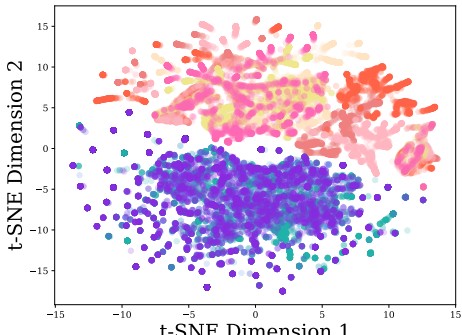 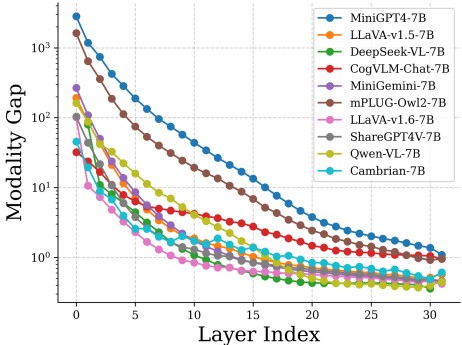

Figure 2: **Current LVLMs show obvious modality gap in the shallow layers.** (**Left**) The t-SNE visualization depicts the significant gap between vision (warm colors) and text (cool colors) tokens at LLaVA-v1.5's embedding space, where we select six types of images (from DocVQA (Mathew et al. (2021)), ChartQA (Masry et al. (2022)), InfoVQA (Mathew et al. (2022)) and ShareGPT4V (Chen et al. (2023))) and three types of text data (from CNN News, Daily Mail (Nallapati et al. (2016)) and Code Search Net (Husain et al. (2019)). (**Right**) The modality gap in different layers of LVLM's language model, which is obtained during computing MIR. For most of LVLMs, the first several layers still strive to narrow the modality gap util the middle layers achieve the alignment.

# 1    INTRODUCTION

In the past two years, the exploration of LVLMs (Liu et al. (2024c); Zhu et al. (2023); Dai et al. (2023)) has exploded from multiple aspects, showcasing amazing usages in many areas. Putting aside the diversity among these explorations, most LVLMs follow a dual-stage paradigm that uses a pre-training stage to align the modality, followed by a supervised fine-tuning (SFT) stage to enable diverse multi-modal capabilities. With the progress of LVLMs, the pre-training stage evolved from lightweight alignment with few image-caption pairs toward deep modality integration with large-scale diverse multi-modal data, playing a more critical role in building capable LVLMs.

Despite the rapid evolution, how to qualify the LVLMs after pre-training is still an under-explored problem, which hinders the pre-training study from being more controllable and reliable. Benchmark results after supervised fine-tuning is the commonly used solution in recent studies, while the SFT stage introduces non-neglectable computation overhead and complex the process.

Another naive idea is to borrow the metrics from the Large Language Models (LLMs) pre-training (Zhao et al. (2023)) and use the loss value, perplexity, or in-context evaluation results to evaluate it. However, as shown in Figure 1, we empirically find that these metrics are neither stable nor reliable in the multi-modal area (Yin et al. (2023); Bai et al. (2023); Chen et al. (2024d); McKinzie et al. (2024)). Specifically, we conduct a controllable experiment about pre-training data quantity to study the relation between these metrics and the benchmark results after supervised fine-tuning. The training loss and perplexity keep decreasing with training data quantity increasing, while the model performance is already saturated with less than 1M data. When it comes to in-context evaluation, its performance shows irregular jitter with varing training data, leading to meaningless indication.

We argue the in-effectiveness of the aforementioned metrics is based on the pre-training target difference between LLMs and LVLMs. The former learns to model the language while the latter focuses on closing the domain gap between two modalities. We sample images and texts from multiple sources and visualize their distribution by the embedding layer feature of the LLaVA-v1.5 (Liu et al. (2024a)). As shown in Figure 2(a), despite the content diversity within images or texts, they show a relatively uniform distribution within each modality and a clear distribution gap between modalities. We further calculate the layer-wise modality gap on several leading LVLMs (Lu et al. (2024); Wang et al. (2023); Ye et al. (2024); Liu et al. (2024b); Tong et al. (2024)) in Figure 2(b), and we have a consistent observation that the gap is closed with the layer increasing, indicating that the LVLMs learn to align the distributions to understand the newly introduced image modality.

Inspired by this, we introduce the Modality Integration Rate (MIR), which measures the pre-training quality of LVLMs from the distribution distance perspective. The MIR enjoys several advantages:

1) **Effective** in representing LVLM pre-training quality, exhibiting a strong correlation with post-SFT benchmark performance. MIR naturally converges during pre-training, offering a useful indicator to aid in identifying critical points during pre-training, such as performance saturation. For instance, as shown in Figure 1 and 6, Table 1, it can effectively identify the saturation point when improving the pre-training data scale or detailedness, making it a reliable tool for early stopping, especially when training on large-scale but homogeneous data.

2) **Robust** across diverse types of image-text pairings used in the evaluation. Benefiting from its distribution perspective design, MIR reflects the distribution distance and be robust toward the specific evaluation data sample. We experimentally find MIR is stable regardless of the input type, including different vision and text contents (e.g., natural or document images, news or mail text), and even irrelevant image-text pairs. This robustness extends to its stability in handling different conversation templates, and ensuring reliability even in the face of overfitting during training.

3) **Generalized** across various training recipes, strategies, and module designs, offering flexibility in evaluating diverse settings or configurations of LVLM pre-training. Whether adjusting training recipes, altering architectural choices, or unlocking specific model modules, MIR helps provide valuable insights into how these variations impact on the quality of cross-modal alignment. For example, MIR offers insight into unlocking strategies in LVLM pre-training, particularly highlighting that unlocking both the projector and language model simultaneously can boost the cross-modal alignment and prevent early saturation when scaling the pre-training data.

Ultimately, MIR proves to be a versatile and consistent metric, facilitating broader LVLM pre-training optimizations. Based on the design of MIR, we further propose **MoCa**, a lightweight and learnable calibration module for each layer's vision tokens, designed to enhance their alignment with text tokens. MoCa achieves the obviously low MIRs on both LLaVA-v1.5 and Mini-Gemini, with +1.5% and +0.9% average gains on post-SFT benchmark performance.

## 2 METHOD

In this section, we give the detailed definition about modality integration rate (MIR) and investigate its basic properties including *input-agnostic*, *training convergence*, and *robustness to overfitting*.

### 2.1 MODALITY INTEGRATION RATE

**Problem Definition.** We aim to establish an effective metric for quantifying the cross-modal alignment quality of LVLM pretraining. This metric should accurately reflect the impact of various pretraining configurations (such as data, strategies, recipes, and architectural choices) on model performance, without requiring subsequent supervised fine-tuning evaluation on benchmarks. Also, it should be applicable across different LVLM architectures.

**Metric Overview.** Consider a pretrained LVLM $\mathcal{M} = (\mathcal{E}, \mathcal{P}, \mathcal{D})$ where $\mathcal{E}$ is the vision encoder, $\mathcal{P}$ is the vision-language projector and $\mathcal{D} = (\mathcal{D}_t, \mathcal{F})$ represents the language model that consists of tokenizer $\mathcal{D}_t$ and $K$-layer transformer decoder $\mathcal{F}$. We input a set of image-text pairs $(\mathcal{V}, \mathcal{T}) = (\{v_n\}_{n=1}^N, \{t_n\}_{n=1}^N)$ to the model, obtaining vision tokens $f_k^{v_n}$ and text tokens $f_k^{t_n}$ from the first $k$ layers $\mathcal{F}_k$ of the language decoder $\mathcal{F}$, i.e., for the $n^{th}$ sample:

$$f_k^{v_n}, f_k^{t_n} = \mathcal{F}_k(\mathcal{P}(\mathcal{E}(v_n)), \mathcal{D}_t(t_n)), \tag{1}$$

where $f_k^{v_n} \in \mathbb{R}^{r_n \times d}, f_k^{t_n} \in \mathbb{R}^{s_n \times d}$ represent the vision and text token representations respectively, from the $k^{th}$ layer output of the $n^{th}$ sample, with $r_n, s_n$ as the number of vision/text tokens and $d$ as the dimension of hidden states. To compute the global domain gap between the two modalities, we further concatenate the vision and text tokens of all image-text pairs at the first dimension, deriving $f_k^v \in \mathbb{R}^{r \times d}, f_k^t \in \mathbb{R}^{s \times d}$ where $r = \sum_{n=1}^N r_n$ and $s = \sum_{s=1}^N s_n$. Besides, we define $f_{k,i}^v \in \mathbb{R}^{1 \times d}$ as the $i^{th}$ vision token in $f_k^v$ and $f_{k,j}^t \in \mathbb{R}^{1 \times d}$ as the $j^{th}$ text token in $f_k^t$.

Our objective is to measure the modality gap between vision tokens $f_k^v$ and text tokens $f_k^t$ at each layer. Given the typical discrepancy in the number of vision and text tokens, i.e., $r \neq s$, we leverage

Fréchet Inception Distance (FID) (Heusel et al. (2017)) compute the domain divergence between these token representations.

**Text-Centric Normalization.** FID is sensitive to the absolute values of features, which is problematic for directly applying FID to evaluate modality gaps across layers, since deeper layers tend to exhibit larger token magnitudes (as indicated by increasing $\ell_2$-norm values) in LVLMs.

A naive solution would be to normalize all tokens to a common scale and compute the distance like cosine similarity. However, it overlooks the significant absolute value disparity between vision and text tokens, which is particularly obvious in the shallower layers. While RMSNorm within each transformer block significantly reduces this disparity, skip connections partially retain its influence, altering the direction of token representations. To address this, we adopt a text-centric normalization that preserves the absolute value differences between vision and text tokens, while neutralizing the effect and enabling cross-layer FID comparison reasonable. Specifically, we first perform $\ell_2$ normalization for text tokens of each layer, deriving a scaling factor $\alpha$ as (Ablation is in appendix):

$$\alpha_k = \frac{s}{\sum_{j=1}^{s} ||f_{k,j}^t||_2}. \tag{2}$$

By multiplying text tokens with the scaling factor $\alpha_k$, the average $\ell_2$-norm of the text tokens is normalized to 1. Thereby, we equally scale both vision and text tokens with the factor $\alpha_k$, to maintain the absolute value differences between vision and text tokens and facilitate more accurate cross-layer comparisons. During implementation, we observe occasional outliers in both vision and text tokens, characterized by unusually high $\ell_2$-norm values. To address this, we apply an outlier removal function $\omega(\cdot)$ based on "$3\sigma$" principle, focusing on the majority of token representations.

**Modality Integration Rate.** After the scaling and outlier removal, we calculate the FID between vision and text tokens at each layer to quantify the modality gap, then aggregate this across all layers to derive the Modality Integration Rate (MIR), i.e.,

$$\text{MIR} = \log \sum_k \text{FID}(\omega(\alpha_k f_k^v), \omega(\alpha_k f_k^t))$$
$$= \log \sum_k \left[ ||\mu_{v,k} - \mu_{t,k}||^2 + \text{Tr}(\Sigma_{v,k} + \Sigma_{t,k} - 2(\Sigma_{v,k}\Sigma_{t,k})^{1/2}) \right] \tag{3}$$

where $\mu_{v,k} = \frac{\sum_i \omega(\alpha_k f_{k,i}^v)}{r'}$, $\mu_{t,k} = \frac{\sum_j \omega(\alpha_k f_{k,j}^t)}{s'}$ are the mean value of the processed vision tokens $\omega(\alpha_k f_k^v)$ and text tokens $\omega(\alpha_k f_k^t)$ at the $k^{th}$ language model layer. $\Sigma_{v,k}$ and $\Sigma_{t,k}$ are the corresponding covariance matrices of $\omega(\alpha_k f_k^v)$ and $\omega(\alpha_k f_k^t)$, which can be formalized by

$$\Sigma_{v,k} = \frac{(\omega(\alpha_k f_k^v) - \mu_{v,k})^\top (\omega(\alpha_k f_k^v) - \mu_{v,k})}{r' - 1}, \quad \Sigma_{v,k} = \frac{(\omega(\alpha_k f_k^t) - \mu_{t,k})^\top (\omega(\alpha_k f_k^t) - \mu_{t,k})}{s' - 1}. \tag{4}$$

To compute the matrix square root term $(\Sigma_{v,k}\Sigma_{t,k})^{1/2}$ in Eq. (3), we typically face high computational costs due to the large matrix dimensions in LVLMs. Therefore, we further provide a solution using Newton-Schulz iteration to approximate the square root, significantly accelerating the process and meeting the practical needs of training indicators. Empirically, this approximation introduces minimal impact on the overall MIR value, with errors generally remaining below 1%.

## 2.2 INVARIANCE TO DIVERSE INPUTS

As a metric that requires some image-text pair data as inputs, MIR is generally input-agnostic for different types of inputs, no matter what types of the images or question-answer pairs, what conversation templates are used, whether the model has seen the data during pretraining, and whether the text is relevant with the image. This property is quite necessary for the practical application of MIR since we can easily or randomly choose some samples for validation to compute the MIR without any bias from data types or sources. This property also proves the proposed MIR exactly reflects the domain gap between the vision and text representations for a particular model, i.e., it should be specific to a particular model but not sensitive to different types of image-text inputs.

Here we present several kinds of scenarios to show MIR's invariance to diverse inputs, with the pretrained model in LLaVA-v1.5 7B:

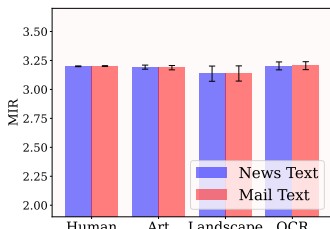 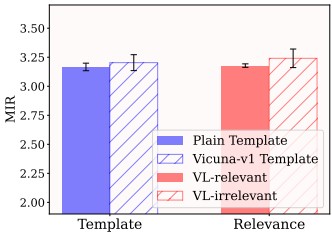 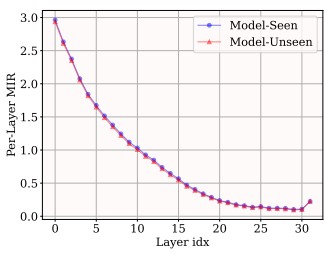

| (a) Visual/Text Contents | (b) Templates & Relevance | (c) Seen *v.s.* Unseen |
| --- | --- | --- |

Figure 3: **MIR is robust to diverse kinds of data.** We compute MIR on pre-trained LLaVA-v1.5 7B model with various inputs to verify whether it is input-agnostic. In (a), we select four kinds of visual contents (human, art, landscape and ocr images) and two kinds of text contents (news, mail text). In (b), we compare the MIR computed on the plain/vicuna v1 conversation template, and image-text relevant/irrelevant pairs. In (c), we select the image-text pairs in pre-training data as "Model Seen" data and unseen pairs as "Model Unseen" data, to depict the per-layer MIR.

**Different contents.** We typically target for four different vision domains (including human, art, landscape, ocr text images) and two language domains (including news text and mail text). For each type of combination, we randomly sample $N = 100$ image-text pairs, where the data is selected from VQA datasets (such as COCO (Lin et al. (2014)), ShareGPT4V (Chen et al. (2023)), ChartVQA (Masry et al. (2022)) and DocVQA (Mathew et al. (2021))) and text datasets (e.g., CNN/DM (Nallapati et al. (2016))) to compute the MIR scores, respectively. Figure 3 (a) shows when we use very different types of visual/text contents to compute MIR, the values are relatively consistent and insensitive to the diverse inputs, indicating its robustness and ability to measure the domain divergence.

**Conversation templates.** Most of LVLMs use their particular template inherited from LLM to support their instruction-following ability, where LLaVA-v1.5 adopts the vicuna v1 template by default after SFT. We try two cases, i.e., with the template and without the template, to compute MIR using the input images from TextVQA (Singh et al. (2019)) validation set (LLaVA has never seen) and text data from CNN/DM. From Figure 3 (b), it is clearly that the introduction of template has relative little impact on the computation of MIR.

**Relevance between image and text.** Here we explore whether the MIR value can be influenced by the correspondence between the images and text selected for computation. We select same 100 images from COCO and prepare two types of text, one is the vanilla caption, the other is irrelevant text in CNN/DM (where we truncate and keep the same number of text tokens). From Figure 3 (b), we surprisingly find that the MIR scores keep similar under the two kinds of inputs. It indicates MIR focuses on the domain divergence between the modalities rather than specific content differences.

**Seen *v.s.* Unseen.** For practical usage, MIR is expected to be invariant whether the input samples for evaluation are involved in the pre-training data, or else the MIR is not reliable to truly show the quality of cross-modal alignment. To this end, we conduct a comparison regarding using model-seen and model-unseen data from COCO to compute the MIR score, respectively. From Figure 3 (c), we can obtain that the two curves are highly overlapped, thus MIR is relatively robust in this case.

By default, unless we specially mentioned, all of MIR computation in this paper uses random $N = 100$ (Ablation is in appendix) images from TextVQA validation set and text data from CNN/DM.

## 2.3 TRAINING CONVERGENCE

In addition to being invariant to diverse inputs, a good pre-training metric should exhibit clear convergence behavior, similar to training loss. Ideally, it should show a sharp decline in the early stages and gradually approach an optimal point, with very slow improvements thereafter. To explore the convergence of MIR, we followed the vanilla pre-training setup of the LLaVA-v1.5 7B model and analyze the relationship among model performance (measured the post-SFT benchmark performance[1]), training loss, and MIR. Figure 4 reveal that MIR demonstrates similar convergence properties to training loss. In particular, both metrics sharply decrease in the early pre-training steps

---

[1] In this paper, we adopt the average score on 7 popular multi-modal benchmarks as the post-SFT model performance, including MMStar, MMBench, MMBench-cn, SEED-Img, TextVQA, ScienceQA-Img, and GQA.

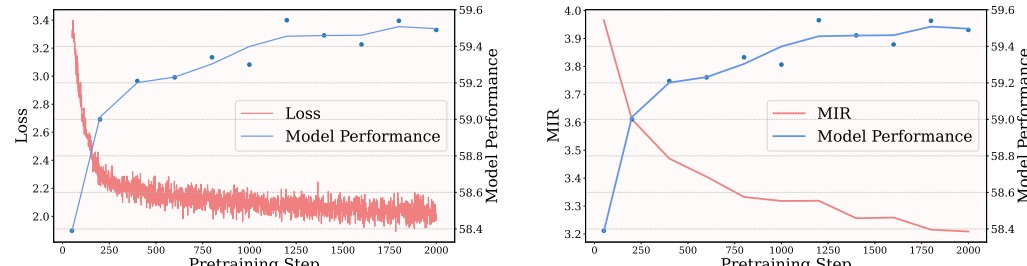

Figure 4: **MIR exhibits the similar convergence properties with training loss, closely corresponds with post-SFT model performance.** We pre-train LLaVA-v1.5 7B model with its vanilla setting and report the training loss, MIR, and post-SFT performance on 7 LVLM benchmarks.

and stabilize over time. Moreover, MIR closely corresponds with post-SFT model performance, making it a strong indicator of effective pre-training.

This convergence also reflects the alignment between vision and language tokens throughout the language model layers, indicating the cross-modal alignment's gradually stabilizing during pre-training. Additionally, MIR's ability to track convergence provides practical value. By using MIR as a pre-training monitor, we can identify when the model has reached sufficient cross-modal alignment, allowing for early stopping and reducing unnecessary training costs.

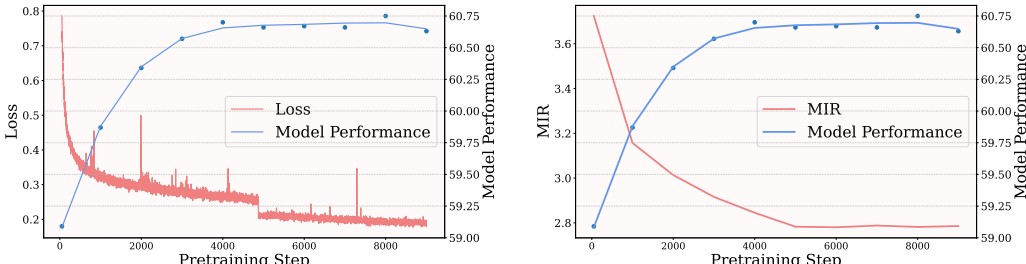

Figure 5: **MIR is robust toward overfitting.** We conduct 2-epoch pre-training based on the settings of ShareGPT4V 7B model, and report training loss, MIR, and post-SFT model performance (averaged on 7 LVLM benchmarks) at each steps. The training loss shows a sharp drop at the beginning of the second epoch while the model performance does not. It shows that MIR is more consistent with the post-SFT model performance than training.

## 2.4 ROBUSTNESS AGAINST MODEL OVERFITTING

Though the training loss can converge at the first epoch, it usually shows a sharp drop at the beginning of the second epoch (Figure 5), due to the overfitting especially when unlocking both the vision encoder and the language model during LVLM pre-training. This drop in loss, however, does not correlate with a significant improvement in model performance, indicating that the loss metric may not accurately reflect the pre-training quality at this stage.

To explore the robustness of MIR in face of such model overfitting, we conduct the evaluation on the stronger baseline ShareGPT4V, which curates around 1.2M detailed image captions for pre-training and unlocks both latter half vision encoder and the whole LLM for better comprehending the detailed semantics. Two epochs of pre-training are performed, and both MIR, training loss, and post-SFT model performance (averaged across 7 benchmarks) are recorded at each step. From Figure 5, it is evident that while the training loss drops significantly at the start of the second epoch, the model performance remains stable after the convergence reached during the first epoch. This lack of correlation highlights how training loss fails to serve as a reliable indicator of performance during overfitting scenarios. In contrast, MIR more closely aligns with model performance, maintaining convergence from the end of the first epoch without drastic fluctuations. This result indicates the robustness of MIR in face of the model overfitting, exhibiting a more reliable indicator than training loss for monitoring the training states of LVLMs.

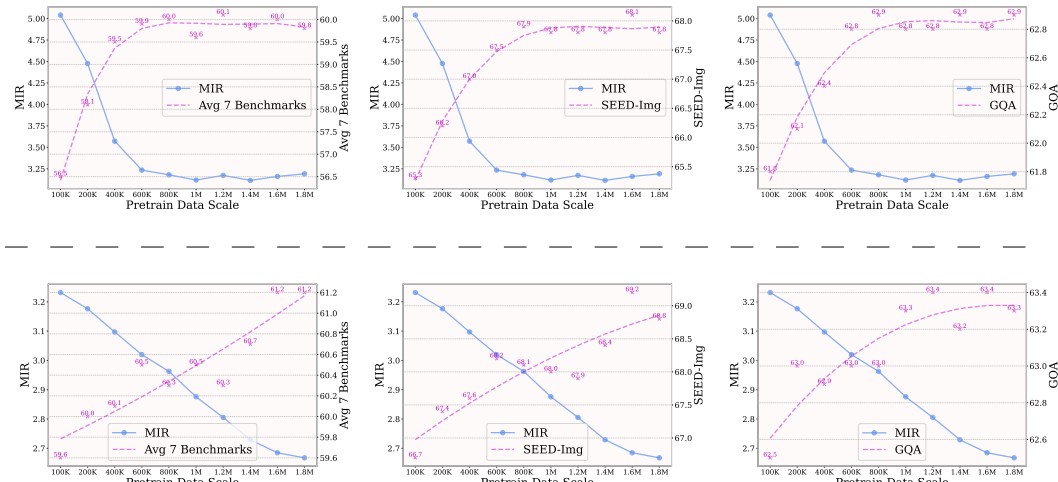

Figure 6: **MIR as an effective evaluator when scaling pre-training data.** We use ALLaVA and ShareGPT4V-PT as ~1.8M per-training data and train each model with different scale of data. **The first row** showcases the results of vanilla LLaVA-v1.5 7B model, where only the MLP projector is trained. **The second row** showcases the results of ShareGPT4V, where the latter half of ViT and the whole LLM are also trained. MIR well aligns with the trend of the post-SFT model performance.

## 3 EXPERIMENT

In this section, we explore several key applications of MIR, proving how it helps optimize configurations for LVLM pre-training. We focus on four scenarios: 1) Unveiling the performance upper bound when scaling pre-training data; 2) Evaluating the impact of text detailedness on the quality of LVLM pre-training; 3) Exploring the potential of MIR to select the optimal training recipes or strategies; 4) Verifying the effectiveness of different module designs for LVLM pre-training..

Our experiments involve three fully open-sourced LVLMs: LLaVA-v1.5, ShareGPT4V, and Mini-Gemini (Li et al. (2024)), focusing on their 7B model variants. For evaluation, we selected 9 popular multi-modal benchmarks, including MMStar (Chen et al. (2024b)), MME (Fu et al. (2023)), MM-Bench (Liu et al. (2023)), MMBench-cn, SEED-Img (Li et al. (2023a)), TextVQA (Singh et al. (2019)), ScienceQA-Img (Lu et al. (2022)), POPE (Li et al. (2023c)), GQA (Hudson & Manning (2019)) and MM-Vet (Yu et al. (2023)). These benchmarks comprehensively test both coarse- and fine-grained capabilities of LVLMs, providing a holistic view of performance across various tasks.

### 3.1 SCALING PRE-TRAINING DATA IN LVLMS

Typically, increasing the amount of training data improves model performance and generalization. Here, we demonstrate the effectiveness of MIR in measuring pre-training quality when scaling the amount of pre-training data. We use LLaVA-v1.5 7B as the base model, and curate two datasets, ALLaVA and ShareGPT4V-PT, comprising ~1.8M image-text pairs. We then pre-trained multiple models using different subsets of this dataset to explore the relationship between pre-training data scale and model performance, with MIR serving as the evaluation metric.

Two different pre-training strategies are employed to investigate this scaling law: *1) Vanilla LLaVA-v1.5 setting:* Only the MLP projector is trained while the rest of the model is frozen, with the learning rate of 1e-3. *2) Vanilla ShareGPT4V setting:* We unlock the latter half of the vision encoder as well as the entire LLM, with a learning rate of 2e-5. After pre-training, we further equally apply supervised fine-tuning to all models on LLaVA's 665K SFT data, using a learning rate of 2e-5. Post-SFT model performance is evaluated across 7 benchmarks to verify the relationship between MIR and pre-training quality, showcasing the effectiveness of MIR as an indicator.

As shown in Figure 6, when only the MLP projector is trained (the first row, LLaVA-v1.5), the post-SFT performance gradually improves but plateaus at 800K~1M data scale, indicating a bottleneck in further enhancing cross-modal alignment. In contrast, when the vision encoder, MLP projector, and LLM are all jointly trained (the second row, ShareGPT4V), continues to increase significantly,

Table 1: **MIR as an effective evaluator when improving the data detailedness.** We truncate the long captions in ALLaVA and ShareGPT4V-PT on the sentence-level, to construct the pre-training data with varing degrees of detailedness. MIR shows strong correlation with post-SFT performance.

| Caption Len | MIR↓ | Average | MMStar | MME | MMB | MMB$^{CN}$ | SEED$^{I}$ | TQA | SQA$^{I}$ | POPE | GQA |
|---|---|---|---|---|---|---|---|---|---|---|---|
| 15.2 | 3.588 | 63.8 | 33.0 | 1488.1 | 65.8 | 59.5 | 66.8 | 57.7 | 69.4 | 86.0 | 62.4 |
| 49.1 | 3.442 | 64.0 | 33.7 | 1500.2 | 65.9 | 58.2 | 67.5 | 59.0 | 68.3 | 86.0 | 62.4 |
| 127.1 | 3.279 | 64.2 | 34.8 | 1472.6 | 66.2 | 58.9 | 67.6 | 59.0 | 68.7 | 85.8 | 62.7 |
| 181.2 | 3.218 | 64.4 | 35.5 | 1491.8 | 65.4 | 57.6 | 67.6 | 59.7 | 69.8 | 86.0 | 63.0 |

even at 1.8M data scale. From these results, we can draw the following insights: 1) MIR serves as an effective indicator when scaling pre-training data. 2) Appropriately unlocking vision encoder or LLM allows for continued improvement for LVLM pre-training on larger-scale data.

## 3.2 IMPROVING DATA DETAILEDNESS IN LVLM PRE-TRAINING

Leveraging the capabilities of MIR, we further explore how varying levels of caption detailedness in pre-training data affect LVLM performance. We select LLaVA-v1.5 as the base model, and use ~1.8M image-text pairs from ALLaVA and ShareGPT4V-PT as the pre-training data. To construct different degrees of caption detailedness of pre-training data, we truncate the original long captions to various lengths one the sentence-level, thus generating captions with varying levels of detailedness. The pre-training procedure follows the default configuration of LLaVA-v1.5, where only the MLP projector is trained with a learning rate of 1e-3.

As shown in Table 1, the results indicate that models trained on more detailed captions tend to have lower MIR values, which correlates with improved post-SFT performance. Notably, when increasing the average caption length from 15.2 to 49.1, the overall model performance improves since the captions can help model comprehend more semantics in the given visual contents. However, when further increasing the average caption length from 127.1 to 181.2, the model's global reasoning ability appears to plateau (as seen in the SEED-Img benchmark), while its fine-grained capabilities, particularly in tasks like TextVQA, continue to show significant improvement.

## 3.3 OPTIMIZING TRAINING RECIPES OR STRATEGIES

Different training strategies and configurations are also a crucial factor for enhancing the quality of LVLM pre-training (Lin et al. (2024)). Here we examine the effectiveness of MIR in optimizing the hyper-parameters and selecting the optimal unlocking strategies during the pre-training phase. To maintain consistency, we use the LLaVA-v1.5 7B model as the base model and standardize the supervised fine-tuning (SFT) across all experiments using LLaVA-v1.5's default setup, which includes 665K GPT-generated SFT data. This allows us to isolate and analyze the impact of various training strategies and configurations on the pre-training stage.

**Training recipes.** We investigate how MIR helps optimize training recipes without requiring additional without further SFT. The baseline setting is LLaVA-v1.5's official pre-training with 558K BLIP-2-generated image-caption pre-training data. We consider the hyper-parameters including the learning rate (LR), warmup ratio and the scheduler type of learning rate decay. From Table 2, we can observe the positive relation between MIR and the post-SFT benchmark performance. Specifically, a lower MIR reflects the effectiveness of different training configurations on pre-training quality, particularly with stable benchmarks like SEED-Img.

**Training strategies.** We explore how MIR can guide the selection of effective unlocking strategies during pre-training. Considering the 558K BLIP-2 generated captions used in vanilla LLaVA-v1.5 are relatively short, we curate ALLaVA and ShareGPT4V-PT as ~1.9M long-caption data for pre-training, following ShareGPT4V's recipe with a learning rate of 2e-5. For unlocking strategies, we focus on unlocking the MLP projector and various parts of the LLM, including LoRA, the first half of LLM, and the entire LLM. As shown in Table 3, unlocking the LLM's parameters significantly reduced MIR and enhanced the model's multi-modal capabilities, indicating a strong correlation between MIR and pretraining quality. These results suggest that, when pre-training on highly detailed image-text data, unlocking the former half of LLM or the entire LLM can significantly improve

Table 2: **MIR as an effective evaluator when optimizing pre-training recipes.** We try different sets of hyper-parameters (learning rate (LR), warmup ratio, and learning rate decay scheduler) to pre-train LLaVA-v1.5 7B models, following its official setting. MIR has strong positive relation with the post-SFT benchmark performance.

| LR | Warmup | LR scheduler | MIR↓ | Average | MMStar | MMB | MMB$^{CN}$ | SEED$^I$ | TQA | SQA$^I$ | GQA |
|----|--------|--------------|------|---------|--------|-----|------------|----------|-----|---------|-----|
| 1e-3 | 3e-2 | cosine | 3.182 | 59.5 | 33.8 | 65.9 | 59.0 | 66.5 | 58.5 | 69.6 | 62.9 |
| 1e-3 | 5e-2 | cosine | 3.043 | 59.6 | 34.2 | 65.7 | 60.0 | 66.9 | 58.5 | 69.0 | 62.7 |
| 1e-3 | 3e-2 | cosine | 3.182 | 59.5 | 33.8 | 65.9 | 59.0 | 66.5 | 58.5 | 69.6 | 62.9 |
| 1e-3 | 3e-2 | linear | 3.171 | 59.5 | 34.9 | 65.6 | 59.5 | 66.7 | 58.4 | 68.3 | 62.8 |
| 3e-3 | 3e-2 | cosine | 3.575 | 58.7 | 33.0 | 64.9 | 59.0 | 65.1 | 58.4 | 68.8 | 62.0 |
| 1e-3 | 3e-2 | cosine | 3.182 | 59.5 | 33.8 | 65.9 | 59.0 | 66.5 | 58.5 | 69.6 | 62.9 |
| 5e-4 | 3e-2 | cosine | 2.990 | 60.0 | 34.6 | 66.8 | 59.5 | 66.8 | 58.8 | 70.3 | 62.9 |
| 3e-4 | 3e-2 | cosine | 2.808 | 60.0 | 35.3 | 67.1 | 59.8 | 67.0 | 58.6 | 69.5 | 62.8 |

Table 3: **MIR as an effective evaluator when optimizing pre-training strategies.** Using ∼1.9M data from ALLaVA and ShareGPT4V-PT, we try to unlock the MLP projector and carious parts of the LLM (including LoRA (Hu et al. (2021)), the former half of LLM, and the entire LLM), and equally apply SFT on LLaVA's 665K SFT data. "w/ merge" means merging the LoRA weights with the model weights after pre-training, *vice versa*.

| Unlock LLM | MIR↓ | Average | MMStar | MME | MMB | MMB$^{CN}$ | SEED$^I$ | TQA | SQA$^I$ | POPE | GQA |
|------------|------|---------|--------|-----|-----|------------|----------|-----|---------|------|-----|
| - | 3.001 | 64.0 | 33.3 | 1479.6 | 66.7 | 59.5 | 67.1 | 58.8 | 68.8 | 85.5 | 62.6 |
| LoRA w/o merge | 2.735 | 64.3 | 33.4 | 1504.6 | 65.5 | 59.5 | 67.7 | 59.0 | 69.2 | 86.0 | 62.8 |
| LoRA w/ merge | 2.734 | 64.5 | 33.4 | 1502.6 | 66.4 | 60.5 | 67.5 | 58.9 | 69.5 | 86.0 | 62.8 |
| Former Half | 2.705 | 65.9 | 34.1 | 1564.5 | 66.8 | 62.1 | 69.2 | 60.1 | 72.4 | 86.5 | 63.3 |
| All Layers | 2.656 | 65.9 | 36.0 | 1554.3 | 66.3 | 62.2 | 69.0 | 60.9 | 71.7 | 86.1 | 63.2 |

the model's ability to bridge the modality gap between vision and language, facilitating the better downstream performance after SFT.

## 3.4 EXPLORING MODULE DESIGNS IN LVLMS

**Vision-language connector.** The architectural design of vision-language connector in LVLMs is critical, since it plays a role in projecting vision features into the language space and narrowing the modality gap. Previous LVLMs typically adopt two kinds of classical visual-language connectors, i.e., MLP and Q-Former (Li et al. (2023b)). The MLP utilizes several linear layers to map visual tokens into the text space, while the Q-Former leverages cross-attention to absorb instruction-aware information from the visual tokens. Here we leverage the proposed MIR to quantify the effectiveness of different types of the vision-language connector on LVLM training. Following the LLaVA-v1.5 setup, we adopt CLIP-ViT-L/336 (Radford et al. (2021)) as the vision encoder and Vicuna v1.5 as the LLM by default. Using the pre-training and SFT data of LLaVA-v1.5, we first pre-train each type of the vision-language connector with the learning rate of 1e-3, keeping the vision encoder and LLM frozen. Afterward, we unlock the LLM to allow joint training with the vision-language connector during instruction tuning. For a fair comparison between MLP and Q-Former, we initialize Q-Former using a BERT-Base checkpoint, without employing any additional warm-up stages to pre-align Q-Former with the vision encoder like BLIP-2.

From Table 4, using an MLP as the vision-language connector significantly outperforms the Q-Former. The 2-layer MLP projector proves to be the optimal choice, achieving the lowest MIR and the highest post-SFT performance. The lower MIR score suggests that MLP facilitates better cross-modal alignment than the Q-Former, allowing it to more effectively comprehend visual information. The positive correlation between MIR and the effectiveness of different vision-language connectors indicates that MIR is a reliable metric for selecting optimal module designs in LVLM training without relying on SFT.

**Learnable Modality Calibration (MoCa).** The observation in Figure 2 shows the base LLMs of LVLMs tend to gradually narrow the modality gap when vision and text tokens are passed through

Table 4: **MIR as an effective evaluator in selecting module designs.** We study the impact of different vision-language connectors on LLaVA-v1.5 training, where we initialize Q-Former with pre-trained BERT-Base (Devlin (2018)) and keep only two stages (pre-training and SFT) for fair comparison. MIR precisely reflects the optimal module design without SFT.

| VL Connector | MIR↓ | Average | MMStar | MME | MMB | MMB$^{CN}$ | SEED$^I$ | TQA | SQA$^I$ | POPE | GQA |
|---|---|---|---|---|---|---|---|---|---|---|---|
| 1-layer MLP | 3.454 | 63.6 | 33.0 | 1439.7 | 65.9 | 59.0 | 66.1 | 58.3 | 69.4 | 86.0 | 62.5 |
| 2-layer MLP | 3.182 | 63.9 | 33.8 | 1446.8 | 65.9 | 59.0 | 66.5 | 58.5 | 69.6 | 86.2 | 62.9 |
| 4-layer MLP | 3.446 | 63.7 | 33.7 | 1436.4 | 65.6 | 59.2 | 66.2 | 58.7 | 69.6 | 86.1 | 62.5 |
| Q-Former | 3.673 | 53.7 | 26.3 | 1175.6 | 56.6 | 49.5 | 51.3 | 45.4 | 67.6 | 76.7 | 51.3 |

Table 5: **The effectiveness of MoCa.** MoCa achieves lower MIRs and +1.5% average benchmark performance on LLaVA-v1.5, as well as +0.9% on Mini-Gemini. Here we report the reproduced results of Mini-Gemini since the data links provided by the official are partially unavailable.

| 7B Model | MIR↓ | Average | MMStar | MME | MMB | MMB$^{CN}$ | SEED$^I$ | TQA | MM-Vet | POPE | GQA |
|---|---|---|---|---|---|---|---|---|---|---|---|
| LLaVA-v1.5 | 3.374 | 59.1 | 30.3 | 1510.7 | 64.3 | 58.3 | 66.1 | 58.2 | 31.1 | 85.9 | 62.0 |
| +MoCa | 3.162 | 60.6 | 36.5 | 1481.0 | 66.8 | 60.0 | 67.0 | 58.7 | 32.2 | 86.9 | 62.8 |
| Mini-Gemini | 2.667 | 62.1 | 34.1 | 1502.8 | 67.5 | 58.2 | 69.5 | 65.2 | 40.8 | 86.1 | 62.3 |
| +MoCa | 2.514 | 63.0 | 35.9 | 1520.5 | 68.3 | 60.2 | 69.6 | 65.6 | 42.9 | 86.5 | 62.4 |

the deeper layers, even though these tokens are not well-aligned when initially fed into the base LLM. It drives us to rethink certain designs in LVLMs that are inherited from LLMs but may be unsuited for promoting cross-modal alignment. One such design is the use of identical normalization for both vision and text tokens at each LLM layer. Since the normalization is pre-trained on language data, it is inherently biased toward text processing, which disrupts vision information and hinders effective cross-modal alignment during training. Therefore, we consider to insert a light-weight learnable module to facilitate such alignment while preserving the language priors in the original LLM normalization modules.

To this end, we propose MoCa, a simple yet effective calibration method specially designed for vision tokens, to help LVLMs automatically adjust the distribution of vision tokens to align more closely with the distribution of text tokens. Specifically, given vision tokens $f_k^v$ and text tokens $f_k^t$ in the hidden states of $k^{th}$ LVLM layer output, we apply a learnable scaling vector $u \in \mathbb{R}^{1 \times d}$ to $f_k^v$ and add it with a learnable shifting vector $v \in \mathbb{R}^{1 \times d}$ before passing it to the next layer, i.e.,

$$\psi(f_k^v) = u \cdot f_k^v + v, \tag{5}$$

where $\psi$ is the learnable calibration module, applied exclusively to vision tokens at the end of each LLM layer. The vector $u, v$ are initialized as an all-ones vector and an all-zeros vector respectively.

We empirically validate MoCa's effectiveness on the 7B models of LLaVA-v1.5 and Mini-Gemini, following their official training configurations. MoCa is integrated into both the pre-training and supervised fine-tuning (SFT) stages. From Table 5, MoCa achieves significantly lower MIR scores on both models, with average post-SFT performance gains of 1.5% for LLaVA-v1.5 and 0.9% for Mini-Gemini. It provides the significant gains on MMStar, demonstrating the learnable vectors effectively bring vision features closer to the distribution of text tokens, ultimately enhancing the model's ability to better comprehend and process visual inputs.

## 4 CONCLUSION

This paper introduces Modality Integration Rate (MIR), a novel and effective metric for evaluating cross-modal alignment during the pre-training of LVLMs. By capturing domain differences between vision and language features across all layers of the language model, MIR provides an effective and reliable measure of pre-training quality compared to traditional metrics like loss or perplexity. It demonstrates its good robustness, generalization ability, and the strong correlation with post-SFT performance, offering valuable insights for optimizing architecture designs and training setup. Complementing MIR, we propose MoCa, a lightweight, learnable calibration module that enhances the alignment of vision and text tokens, ultimately driving better multi-modal comprehension.

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

# A RELATED WORK

## A.1 VISION-LANGUAGE FOUNDATION MODEL.

Vision-Language Models (VLMs) have emerged as a significant advancement in nowadays' multi-modal learning, capable of understanding and generating human-like responses based on visual and textual inputs. Early models like CLIP (Contrastive Language–Image Pre-training) (Radford et al. (2021)) marks a pivotal moment by aligning images and text in a shared embedding space, enabling the strong cross-modal understanding. Following CLIP, models like BLIP (Bootstrapping Language-Image Pre-training) (Li et al. (2022; 2023b)) extends this foundation, enhancing the fusion of vision and language modalities by leveraging more complex pre-training objectives. As the capabilities of Large Language Models (LLMs) (Zhao et al. (2023); Touvron et al. (2023)) progressed, their integration with vision models gave rise to more powerful instruction-following Large Vision-Language Models (LVLMs) (Liu et al. (2024c;a;b); Zhu et al. (2023); Dai et al. (2023); Bai et al. (2023); Zhang et al. (2023); Chen et al. (2024c); Qiao et al. (2024)). Early models such as Flamingo (Alayrac et al. (2022)) and PaLM-E (Driess et al. (2023)), and more recent ones like LLaVA (Liu et al. (2024c)) and Qwen-VL (Bai et al. (2023)), exemplify this trend.

Most LVLMs share three essential components: *the vision encoder*, *the vision-language connector*, and *the language decoder*. *The vision encoder* is responsible for extracting precise features from images, capturing both detailed and abstract visual information. Popular choices include CLIP (Radford et al. (2021)), OpenCLIP (Ilharco et al. (2021)), EVA-CLIP (Sun et al. (2023)), SigLIP (Zhai et al. (2023)) and DINO series (Oquab et al. (2023)), which are designed to provide both coarse-grained and fine-grained visual guidance. *The vision-language connector* plays a critical role in mapping the encoded visual features into a format that can be interpreted by the language model. Common designs include simple MLP projectors and the Q-Former used in BLIP-2, while more advanced solutions, such as the vision abstractor in mPLUG-Owl (Ye et al. (2023)) and QLLaMA in Intern-VL (Chen et al. (2024e)), push the boundaries of cross-modal alignment. *The language decoder* is typically a pre-trained LLM designed to handle large-scale language data, ensuring that the model has robust instruction-following and conversational abilities. However, the central challenge in building a strong LVLM lies in bridging the modality gap between vision and language. The goal is to ensure that the language decoder can process visual tokens as naturally as it does language tokens, enabling smooth and meaningful conversations with multi-modal inputs. This crucial process is typically addressed during the pre-training stage of LVLM development. In this paper, we focus on evaluating and improving cross-modal alignment during the pre-training of LVLMs, a critical step in enhancing their overall performance and ensuring seamless interaction between visual and textual modalities.

## A.2 CROSS-MODAL ALIGNMENT IN LVLMS.

Cross-modal alignment plays a pivotal role in building a strong LVLM that can well support users to input images/videos and the model can understand the multi-modal contents. For the connector module of cross-modal alignment, there are typically three types widely used in current LVLMs: 1) Flamingo-style (Alayrac et al. (2022)). The perceiver resampler projects the vision features into the fixed number of vision tokens, and the language decoder captures the vision information by introducing cross-attention in Gated XATTN-DENSE layer. 2) BLIP-2-style (Li et al. (2023b)). A Q-Former to extract the instruction-aware information from vision tokens through cross-attention and pass the extracted tokens to the language decoder. 3) LLaVA-style (Liu et al. (2024c)). A simple MLP projector directly map the vision tokens into the text embedding space.

Current Large Vision-Language Models (LVLMs) typically undergo a pre-training stage specifically designed for cross-modal alignment. As a result, the quality of the pre-training data and the strategies employed are critical for enhancing this alignment. Early datasets, such as COCO (Lin et al. (2014)), Flickr30k (Plummer et al. (2015)), and LAION-400M (Schuhmann et al. (2021)), focus on short captions describing visual content. More recent datasets like ShareGPT4V (Chen et al. (2023)) and ALLaVA (Chen et al. (2024a)) feature longer captions, aiming to provide richer descriptions to encourage the model to fully utilize the dense information of vision tokens. Besides, some works have shown that incorporating grounding information (Peng et al. (2023)) or dense priors Krishna et al. (2017) in the captions further enhances LVLMs' ability to comprehend

visual inputs. High-quality data plays a key role in improving the cross-modal alignment in LVLMs, driving advancements in multi-modal understanding.

# B  APPENDIX EXPERIMENTS

## B.1  THE NECESSITY OF TEXT-CENTRIC NORMALIZATION IN MIR

The computation of our MIR requires text-centric normalization for both vision tokens and text tokens. This design ensures fairness in cross-layer comparisons of MIR, as FID values are sensitive to the absolute magnitudes of the inputs. To explore this further, we ablated the scaling factor used in MIR computation, and the results are shown below:

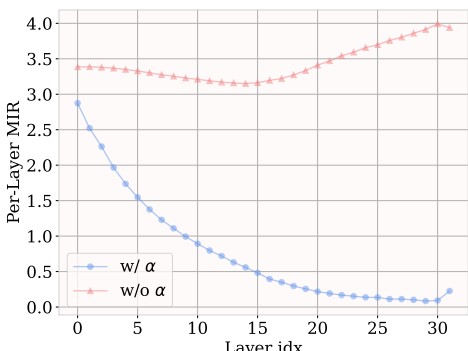

Figure 7: Text-centric normalization is necessary for MIR computation. We ablate the $\alpha$ in MIR and find that it can help MIR to realize the fair cross-layer comparison.

Without text-centric normalization, the MIRs across different layers of the language model exhibit a pattern of first decreasing and then increasing, with the final MIR even higher than that of the first layer. This is counterintuitive because the deepest layer is closest to the language supervision, and the vision/text tokens at that layer should be more tightly aligned. For example, if we attempt to find the closest text embeddings for the vision tokens in the deepest layer across the vocabulary, we will observe much more semantic alignment compared to the vision tokens in the first layer. Therefore, without text-centric normalization, MIRs across layers become incomparable due to differences in absolute values, rendering cross-layer MIR comparisons unfair. Hence, applying text-centric normalization in MIR is essential for meaningful comparisons.

## B.2  IS MIR SENSITIVE TO THE NUMBER OF DATA SAMPLE?

As we clarified in the Method, we use 100 random selected images from TextVQA validation set and text data from CNN/DM for MIR calculation. Hereby, we explore the sensitivity of MIR to the number of data samples. We randomly choose 10 sets of the certain number of data samples to compute MIR for pre-trained LLaVA-v1.5 7B model, reporting the average values and ranges under different data sample numbers.

The results are as below:

Table 6: The mean value of MIR gradually becomes stable with the increase of sample number.

| #Samples | 1 | 5 | 10 | 20 | 50 |
|---|---|---|---|---|---|
| LLaVA-v1.5 7B | 3.380 | 3.358 | 3.377 | 3.379 | 3.374 |
| #Samples | 100 | 200 | 500 | 800 | 1000 |
| LLaVA-v1.5 7B | 3.375 | 3.376 | 3.376 | 3.376 | 3.376 |

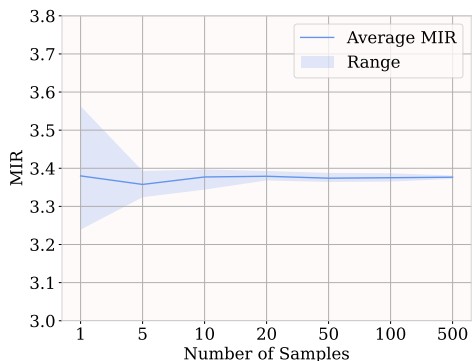

Figure 8: The fluctuation amplitude of MIR gradually decreases with the increase of sample number.

It can be concluded that, if we use more than 20 samples to compute MIR, the fluctuation range is relatively small and we just need to compute MIR for one times as the negligible error, instead of computing for multiple times to get average value. Overall, MIR is relatively robust to the number of data samples, which is effective and reliable when $N \geq 20$.

### B.3 FURTHER DISCUSSION *w.r.t* PERPLEXITY (PPL) IN LVLMS

In Figure 1, we show the PPL is not precise to indicate the pre-training quality. This result is draw from computing PPL on LLaVA-v1.5 7B model that is pre-trained on GPT-style pre-training data (i.e., ALLaVA and ShareGPT4V-PT) and evaluating with the samples selected from ShareGPT4V, which means the training data and the evaluation samples are from the same domain. Here we should argue that PPL is much less reliable when the pre-training data has domain gap with the evaluation samples. To this end, we conduct the experiments on the ~1.2M data by mixing LLaVA's BLIP-2-generated 558K data and ALLaVA, to pre-train LLaVA-v1.5 7B model with different scale of data. Then we follow the same evaluation settings to compute PPL and MIR, here is the comparison.

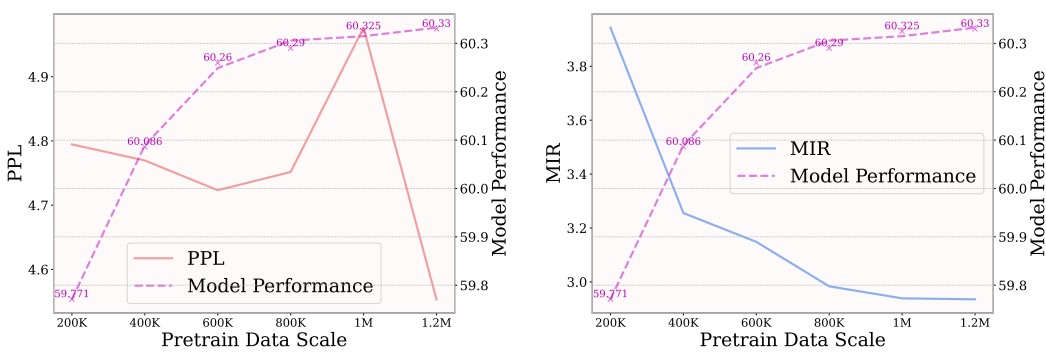

Figure 9: PPL is much less reliable when the pre-training data has domain gap with the evaluation samples.

It indicates that PPL is not appropriate for evaluating the pre-training quality of LVLMs, which is struggling to deal with LVLMs' diverse pre-training data from multiple domains nowadays. In contrast, MIR offers a reliable evaluation for LVLM pre-training without SFT.

### B.4 LARGER LVLMS

We further study the MIRs of LVLMs that have different scale of base LLMs. All of pre-training data and recipes are the same with the official setting of LLaVA-v1.5. The results are as the following:

Table 7: MIR values of LVLMs that have different scale of LLMs.

| Base LLM | Vision Encoder | Projector | Pretrain Data | Epoch | MIR |
|---|---|---|---|---|---|
| Vicuna-13B-v1.5 | CLIP-L/336 | MLP-2x | LCS-558K | 1epoch | 2.583 |
| Vicuna-7B-v1.5 | CLIP-L/336 | MLP-2x | LCS-558K | 1epoch | 3.374 |
| LLaMA-2-13B-Chat | CLIP-L/336 | Linear | LCS-558K | 1epoch | 2.477 |
| LLaMA-2-7B-Chat | CLIP-L/336 | Linear | LCS-558K | 1epoch | 3.699 |

The results above show that the 13B base LLM achieves a lower MIR than the 7B base LLM, indicating that the larger, well-trained LLM has a stronger capability to narrow the modality gap in the shallow layers (as MIR is heavily influenced by the larger modality gap in the shallow layers of the language model). This is also consistent with the improved post-SFT multi-modal performance of the 13B model.

