# OpenReview forum: "Deciphering Cross-Modal Alignment in Large Vision-Language Models with Modality Integration Rate"
_ICLR.cc/2025/Conference — ICLR 2025 Conference Withdrawn Submission_

### Official Review · Reviewer_37mx · 2024-10-22

**Soundness:** 3
**Presentation:** 3
**Contribution:** 3
**Rating:** 5
**Confidence:** 3

**Summary:**

This paper introduces a novel metric, the Modality Integration Rate (MIR), to assess cross-modal alignment quality in large-scale Vision Language Models (LVLMs) during their pre-training phase. Through extensive experiments, the authors demonstrate the utility and effectiveness of MIR across a variety of pre-training configurations, thereby proving its practicality and efficacy in evaluating and optimizing LVLM pre-training processes.

**Strengths:**

1. The introduction of MIR addresses the shortcomings of existing pre-training evaluation metrics, providing a fresh perspective on model assessment.
2. The extensive experimental validation of MIR confirms its stability and predictive power, enhancing the credibility of the research.
3. The study not only discusses theoretical aspects but also provides concrete application examples, offering valuable guidance for practical model training and optimization.

**Weaknesses:**

1. While the paper demonstrates the practicality of the MIR metric through experimental validation, the discussion on its theoretical basis may be lacking. The absence of a deep mathematical analysis of the metric's nature and influencing factors might hinder understanding of its theoretical support and conditions for applicability.
2. The verification of MIR primarily utilizes standard, large-scale visual language datasets, such as ALLaVA and ShareGPT4V-PT. These datasets might share similar characteristics, limiting the indicator's demonstrated generalizability. Further verification experiments are necessary to establish broader applicability.
3. The paper includes multiple experiments to verify MIR's effectiveness but does not provide a sensitivity analysis of MIR values under various parameter settings. Expanding the analysis to include variations in input data characteristics, model configurations, or training strategies could help evaluate its stability and reliability.
4. Although the paper showcases the practicality and effectiveness of MIR, it lacks a direct comparison with other existing evaluation methods. To better highlight MIR's advantages, further quantitative comparative experiments with other modal fusion indicators are recommended.

**Questions:**

1. The paper employs the Fréchet Inception Distance (FID) to calculate the distribution distance between modalities. Is there a specific rationale behind this choice? Are there alternative metrics that could also be suitable?
2. The computational time and resource consumption of MIR are not detailed in the paper. What is the computational complexity of MIR when applied in actual large-scale model training?
3. Has there been consideration for performing a parameter sensitivity analysis on MIR? For instance, how do variations in the number of model layers, the scale of modality pairing data, or different regularization strategies impact MIR values?

---

### Official Review · Reviewer_ATyc · 2024-10-28

**Soundness:** 3
**Presentation:** 3
**Contribution:** 2
**Rating:** 5
**Confidence:** 2

**Summary:**

This paper introduces the Modality Integration Rate (MIR) as an evaluation metric for the pre-training quality of large vision-language models (LVLMs) and proposes a lightweight Modality Calibration (MoCa) module to enhance alignment between visual and textual modalities. The MIR metric demonstrates good robustness and generality, effectively guiding data scale, strategies, and model design during pre-training. However, the paper shows some limitations in terms of innovation and experimental rigor. For instance, the visualization in Figure 2 lacks persuasiveness, and the choice of weights for the mean and covariance in the MIR formula has not been thoroughly explored. Overall, MIR and MoCa provide a new approach for cross-modal alignment in LVLMs, but there is room for improvement in theoretical discussion and experimental validation.

**Strengths:**

1.	Introduction of an Innovative Evaluation Metric (MIR): The paper introduces the Modality Integration Rate (MIR), a novel metric to quantify cross-modal alignment quality during the pre-training stage of large vision-language models (LVLMs). MIR effectively reflects pre-training quality without relying on supervised fine-tuning, filling a gap in pre-training evaluation for LVLMs.
2.	Wide Applicability and Robustness: MIR demonstrates robustness across different types and quantities of data inputs. It adapts well to various modal inputs (such as different types of images and texts) and remains stable in the face of overfitting and changes in data distribution. This makes MIR highly generalizable across different data and model configurations.
3.	Practical Optimization Guidance: The experiments showcase the application of MIR in optimizing data scale, data detail level, training strategies, and model architecture design, providing practical guidance for multi-modal pre-training. MIR helps researchers identify optimal data scale and training strategies during pre-training, thereby improving training efficiency.
4.	Introduction of the MoCa Module for Enhanced Alignment: The paper proposes a lightweight Modality Calibration (MoCa) module that further improves cross-modal alignment by calibrating visual features. MoCa reduces MIR values and enhances multi-modal task performance, offering an efficient solution for cross-modal alignment in LVLMs.
5.	Advantage over Traditional Metrics: Compared to traditional metrics such as loss function, perplexity, and other evaluation metrics, MIR shows stronger indicative power and stability, especially in multi-modal scenarios. The experiments demonstrate that MIR is more effective at capturing the fusion between visual and textual modalities during multi-modal pre-training.

**Weaknesses:**

1.	The paper’s originality appears limited; it is recommended that the author consider either enhancing the contribution or submitting to a general CCFA venue.
2.	The visualization in Figure 2 lacks persuasiveness. Using t-SNE for modality difference visualization appears somewhat redundant. t-SNE requires extensive parameter tuning, and the choice of random data can influence visualization outcomes, making it highly flexible. This flexibility allows almost any pattern to display differences through t-SNE, thus reducing rigor and persuasive impact.
3.	The optimality of the ratios in the MIR formula has not been explored. The MIR formula uses a 1:1 weight ratio of mean distance and covariance to measure modality differences. However, it remains unverified whether this ratio is optimal. Investigating alternative ratios might allow for more effective assessment of pretraining quality. Further exploration of how different weightings affect MIR’s effectiveness in various application contexts is recommended to enhance the indicator’s adaptability and versatility.

**Questions:**

Refer to Weakness.

---

### Official Review · Reviewer_GxLA · 2024-10-30

**Soundness:** 3
**Presentation:** 3
**Contribution:** 2
**Rating:** 3
**Confidence:** 4

**Summary:**

The paper presents a metric called Modality Integration Rate (MIR), designed to evaluate the pre-training quality of Large Vision-Language Models (LVLMs) by measuring the distribution distance between visual and textual features. Unlike traditional metrics like loss or perplexity, MIR is shown to be a more effective and robust indicator of pre-training performance, particularly in aligning vision and language modalities. The paper demonstrates MIR’s utility in assessing training strategies, dataset choices, and model configurations.

**Strengths:**

The evaluation of modality alignment after pre-training LVLMs has not been explored before. The authors highlighted this gap and designed a relatively good metric to address this issue.

**Weaknesses:**

1. The authors merely adapted the FID metric to create the proposed MIR score, which limits the originality of the paper's contribution.
2. FID has recently faced criticism as an evaluation metric. The authors should refer to [1] for more details. Besides, why did they decide to use FID rather than some other metrices, for example, KL-divergence, mutual information, maximum mean discrepancy and so on?
3. The experiments were conducted only on LLaVa-v1.5. To demonstrate MIR's generalizability, the authors should have tested it on more LVLMs. Additionally, the datasets used in the study are too few to prove MIR's effectiveness and generalizability.
4. In my view, the paper's motivation is somewhat lacking. Given the existing metrics for evaluating LVLM performance, introducing a new intermediate metric feels unnecessary.

[1] Jayasumana, Sadeep, et al. "Rethinking fid: Towards a better evaluation metric for image generation." Proceedings of the IEEE/CVF Conference on Computer Vision and Pattern Recognition. 2024.

**Questions:**

Please see Weaknesses above.

**Details Of Ethics Concerns:**

No.

---

> ### Author Response · Authors · 2024-11-13
> **Response to Reviewer GxLA (1/2)**
>
> Thank you for your thoughtful reviews. We appreciate the opportunity to clarify key contributions in our paper: the Modality Integration Rate (**MIR**), a reliable metric for evaluating LVLM pre-training; the learnable Modality Calibration (**MoCa**), a novel lightweight module ot help LVLM narrowg the modality gap during training.
>
> Here we would like to first address some questions to clear up any misunderstandings, which may have arisen from our writing or formulation.
>
> **Q1: What is the necessity of proposing MIR?**
>
> **A1**: It’s essential to clarify MIR’s role in LVLM pre-training. As we highlighted in the Introduction, there has been **no** reliable metric for assessing pre-training quality specific to LVLMs. Traditional evaluation methods borrowed from LLMs, like perplexity, are **ineffective** for evaluating LVLM pre-training (See **Figure 1** in our paper, they can not accurately reflect the post-SFT model performance on downstream benchmarks) due to the different pre-training objective with LLMs. So the common way to evaluate the pre-training quality of LVLM is actually fixing the SFT data/setting to apply SFT on the pre-trained LVLM, then report performance across benchmarks. This approach, however, is **costly** and **time-consuming**. **Therefore, MIR addresses a critical need for a reliable, direct evaluation metric for LVLM pre-training that avoids the need for subsequent SFT.**
>
> **Q2: What is the motivation behind MIR’s design?**
>
> **A2**: Our design in MIR stems from the motivation shown in **Figure 2** of our paper, where we demonstrate that current LVLMs exhibit noticeable modality gaps at shallow layers (including the text embedding space). During training, the shallow layers of the base LLM strives to bridge this gap. This perspective **contrasts** with the common view that the projector (e.g., LLaVA’s MLP projector) can map vision representations directly into the text embedding space. Thus, MIR evaluation is on **two main aspects**:
> - Whether the pre-trained projector can precisely map the vision representations to the similar domain distribution with the language's;
> - Whether the base LLM has the capability to quickly narrow the modality gap between vision and language at the shallow layers.
>
> Our MIR design builds upon the two aspects, with core computation steps that include accumulating the modality gap across all layers, text-centric normalization, and outlier removal.
>
> **Q3: Does MIR simply use FID in its implementation?**
>
> **A3**: We should clearly clarify about the designs and insights about MIR. The method part of our paper may make some misunderstandings for you on the design of MIR. Indeed, MIR **does not simply use FID** but instead involves **three key steps**:
>
> - **Text-Centric Normalization**: MIR has a text-centric normalization for both vision and text tokens at the same LLM layer. It is motivated by the sensitivity of Frechet distance on the absolute value scale (e.g., calculating Frechet distance on (100x, 100y) will obtain the much larger result than calculating Frechet distance on (x, y)), which makes the results not accurate since the absolute values of token representations are gradually larger when the LLM layer goes deeper. This mechanism enables the reliable cross-layer comparison and the reasonable cross-layer accumulation in Step 3.
> - **Outlier Removal**: MIR uses the “3-$\sigma$” principle to remove outliers for both vision and text tokens, recognizing that some outlier tokens contribute little semantically but can skew the distributions [1,2,3]. This step reduces outlier influence, which is crucial as Frechet distance is sensitive to such deviations.
> - **Layer-wise accumulation**: MIR calculates Frechet distances (without Inception encoding) between vision and text tokens at each LLM layer, accumulates these distances, and computes the logarithm of the result for the final MIR score. This cross-layer accumulation, although straightforward, is well-motivated by the observations discussed in **A2**.
>
> We hope these clarifications provide better insight into the motivations and design of MIR, and we look forward to addressing any further questions. We will try to address the concerns raised in **Weakness 2\&3** at the next part of our response.
>
>
> [1] Xiao et al. Efficient Streaming Language Models with Attention Sinks. ICLR, 2024.
>
> [2] Sun et al. Massive Activations in Large Language Models. COLM, 2024.
>
> [3] Gu et al. When Attention Sink Emerges in Language Models: An Empirical View. Arxiv preprint 2410.10781, 2024.

---

### Official Review · Reviewer_Evdn · 2024-11-01

**Soundness:** 4
**Presentation:** 4
**Contribution:** 3
**Rating:** 8
**Confidence:** 3

**Summary:**

This work proposes Modality Intergration Rate (MIR) to evaluate the pre-training quality of LVLMs, which measures the distance between vision and text modalities. The experiments show that MIR is not only robust but also highly related to the pre-training quality, showing a positive relation with the SFT performance. Inspired by the analysis of MIR, the authors further introduce a lightweight module MoCa in SFT to improve multi-modal understanding.

**Strengths:**

- MIR as a new metric is proposed to evaluate the pre-training quality of LVLMs. Sufficient experiments in the paper shows the effectiveness, robustness and generalisability of MIR.
- Based on the MIR design, a lightweight module MoCa is proposed and used in SFT, improving multi-modal understanding.
- The paper is well written, clear and easy to read.

**Weaknesses:**

- As shown in Equ. 3, MIR is calculated by summing the FID of each LLM layer. Therefore, my concern is whether MIR is still comparable for LVLMs with different backbone LLMs, especially those with different depths and sizes. It seems that MIR is less generic than loss and in-context evaluation metrics at this point.

**Questions:**

- In Equ. 3, how is the ω function based on the "3σ" principle implemented? Have the authors examined the characteristics of the outlier tokens?
- How does the MIR change at each layer of the model with MoCa, compared with the one without it? Is there a decrease in per-layer MIR as expected?

---

### Note · Authors · 2024-11-15

I have read and agree with the venue's withdrawal policy on behalf of myself and my co-authors.